# Abundance and Diversification of Repetitive Elements in Decapoda Genomes

**DOI:** 10.3390/genes14081627

**Published:** 2023-08-15

**Authors:** Christelle Rutz, Lena Bonassin, Arnaud Kress, Caterina Francesconi, Ljudevit Luka Boštjančić, Dorine Merlat, Kathrin Theissinger, Odile Lecompte

**Affiliations:** 1Department of Computer Science, ICube, UMR 7357, University of Strasbourg, CNRS, Rue Eugène Boeckel 1, 67000 Strasbourg, France; christelle.rutz@etu.unistra.fr (C.R.); bonassin@unistra.fr (L.B.); akress@unistra.fr (A.K.); luka.bostjancic@senckenberg.de (L.L.B.); dorine.merlat@etu.unistra.fr (D.M.); 2LOEWE Centre for Translational Biodiversity Genomics (LOEWE-TBG), Senckenberg Biodiversity and Climate Research Centre, Georg-Voigt-Str. 14-16, 60325 Frankfurt am Main, Germany; francesconi@uni-landau.de (C.F.); kathrin.theissinger@senckenberg.de (K.T.); 3Department of Molecular Ecology, Institute for Environmental Sciences, Rhineland-Palatinate Technical University Kaiserslautern Landau, Fortstr. 7, 76829 Landau, Germany

**Keywords:** transposable elements, satellite DNA, Crustacea, annotation, evolution, genome size, library

## Abstract

Repetitive elements are a major component of DNA sequences due to their ability to propagate through the genome. Characterization of Metazoan repetitive profiles is improving; however, current pipelines fail to identify a significant proportion of divergent repeats in non-model organisms. The Decapoda order, for which repeat content analyses are largely lacking, is characterized by extremely variable genome sizes that suggest an important presence of repetitive elements. Here, we developed a new standardized pipeline to annotate repetitive elements in non-model organisms, which we applied to twenty Decapoda and six other Crustacea genomes. Using this new tool, we identified 10% more repetitive elements than standard pipelines. Repetitive elements were more abundant in Decapoda species than in other Crustacea, with a very large number of highly repeated satellite DNA families. Moreover, we demonstrated a high correlation between assembly size and transposable elements and different repeat dynamics between Dendrobranchiata and Reptantia. The patterns of repetitive elements largely reflect the phylogenetic relationships of Decapoda and the distinct evolutionary trajectories within Crustacea. In summary, our results highlight the impact of repetitive elements on genome evolution in Decapoda and the value of our novel annotation pipeline, which will provide a baseline for future comparative analyses.

## 1. Introduction

With over 15,000 living species, Decapoda represents a diverse order of Crustacea that includes lobsters, crayfish, crabs, prawns, and shrimps [1]. They are a crucial component of marine and freshwater ecosystems [2,3]. The Decapoda order originated around 455 million years ago, in the Late Ordovician, and is divided into two suborders: the Dendrobranchiata (commonly known as prawns) and the Pleocyemata. The latter encompasses Caridea (swimming shrimps) and a crawling/walking group called Reptantia that consists of Achelata (spiny lobsters), Astacidea (true lobsters and crayfish), Anomura (hermit crabs), and Brachyura (short-tailed crabs) [4].

Decapoda are characterized by highly variable genome sizes. According to the Animal Genome Size Database (https://www.genomesize.com, accessed on 17 May 2022), genome size estimates range from 2.3 Gb for *Penaeus duorarum* to 5.1 Gb for *Aristaeomorpha foliacea* in the Dendrobranchiata suborder. In Pleocyemata, particularly in the Caridea infraorder, genome size variations are even more striking, with estimates ranging from 3.2 Gb for *Antecaridina* sp. to 40 Gb for *Sclerocrangon ferox*. Freshwater crayfish (Astacidea infraorder) also display substantial genome size variations, ranging from 2 to 6 Gb in Cambaridae and Parastacidae families. Recent genome size estimates for the noble crayfish *Astacus astacus* and the narrow-clawed crayfish *Pontastacus leptodactylus*, both representatives of the Astacidae family, reach 17 Gb (K. Theissinger, unpublished results) and 18.7 Gb [5], respectively. Decapoda also displays high variation in the number of chromosomes. The number of chromosomes in the Dendrobranchiata suborder is mainly at a 2n of 88 (reviewed in [6,7]), while this number can explode in Pleocyemata species to a 2n of 376 for the Astacidea *Pacifastacus leniusculus* [8,9].

Variations in genome sizes are usually attributed to the presence of repetitive elements (REs), which can represent the major part of the genome in some eukaryotic species [10]. A high proportion of REs can greatly complicate genome sequencing and can lead to fragmented and incomplete assemblies [11,12,13]. This may explain the notorious difficulties encountered in the sequencing of large Decapoda genomes, with only eight assemblies available at the chromosome level. To date, the relationship between the genome size and repeat content, and the impact of REs on genome evolution, remain poorly studied in Crustacea.

The role of REs can be diverse (reviewed in [14]). They can affect transcription and regulation at transcriptional and post-transcriptional levels. Through their ability to act as signals to locate and process information stored in coding sequences, they can influence damage repair, DNA restructuring, chromatin and nuclear organization, and cell division. REs can be classified into two types: tandem repeats (satellite DNA, satDNA) and transposable elements, TEs, also known as interspersed repeats [15].

SatDNAs consist of tandemly repeated patterns of nucleotides, called repeat units (monomers) [16]. Different satDNA families are present in the genome, with usually only one or a few predominant families [17,18,19,20]. SatDNAs can have specific roles in gene and genome regulation, such as chromosome organization, pairing, and segregation formation of the centromere locus [21,22], in epigenetic regulation of heterochromatin establishment, and modulation of gene expression in response to stress [23,24]. In Crustacea, some SatDNA transcripts can have an impact on the inter-molt stage [25]. Despite their importance, the distribution patterns, percentage, and copy number of satDNAs are not yet fully explored in Crustacea.

Transposable elements (TEs) are mobile elements known to participate in DNA replication and cause gene rearrangements that can confer new functional properties [26,27,28,29]. Deletions, duplications, and inversions can be caused by recombination events between homologous regions dispersed by related TEs at distant genomic positions. When they are inserted into genes or coding regions, TEs can alter gene expression and may produce deleterious effects, such as diseases, or neutral effects on the host [28,30,31,32]. Organisms living in challenging environmental conditions can have more TEs in their genome, increasing genome plasticity to respond to stress factors [33]. TEs can be divided into two classes based on their replication mechanisms: Class I elements transpose with RNA-mediated mechanisms (retrotransposons), while in Class II the transposition mode is DNA-based (DNA transposons) [34,35,36,37]. In Class I, LTR retrotransposons and Penelope-like elements are characterized by Long Terminal Repeat (LTR). DIRS are bound by direct or inverted repeats. Finally, LINEs (long interspersed nuclear elements) and SINEs (short interspersed nuclear elements) are retrotransposons that do not have terminal repeats but a polyA tail at the 3’ end. Unlike LINEs, SINEs evolved from non-coding RNA genes and are non-autonomous. Class II can be divided into two subclasses. Subclass 1 includes TIR and Crypton elements, while subclass 2 includes Helitrons and Mavericks. Apart from SINEs, most TEs encode proteins that are necessary for their transposition in an autonomous way. However, accumulation of mutations can lead to incomplete versions of TEs that no longer encode transposition enzymes. The identification of these truncated alternatives represents a particular challenge for automated annotation pipelines.

Currently, there are several pipelines available for annotation of REs. The most commonly used tools are RepeatModeler2 [38] and RepeatMasker [39]. However, a wide variety of additional tools have been developed, such as RECON [40], RepeatScout [41] and LtrHarvest/Ltr_retriever [42], REPET [43], RepeatExplorer [44] (based on paired-end reads). The availability of multiple tools highlights the lack of a standardized protocol, making it impossible to directly compare the RE composition between different genomes based solely on the literature. Moreover, current pipeline annotations of REs fail to identify a significant portion of divergent repeats in non-model organisms. To address these limitations, we designed a standardized protocol for RE annotation that encompasses both TEs and satDNAs. This pipeline was used to establish the RE landscape of twenty Decapoda and six other Crustacea, enabling an objective comparison of the Decapoda repeatomes in terms of abundance, composition, and evolutionary dynamics. Our standardized approach allowed us to assess the contribution of REs to the evolution of the enigmatic Decapoda genomes. Furthermore, we explored the possibility of using the REs as reliable phylogenetic markers for Decapoda. Lastly, this study also provides a new library of REs in Decapoda genomes that extends the existing databases and can be used for future analyses.

## 2. Materials and Methods

### 2.1. Genomic Datasets

Available assemblies for Decapoda species were downloaded from NCBI GenBank and RefSeq (last accessed 16 February 2022). Contig and scaffold N50 are useful values to estimate the contiguity of the genome by indicating the length of the shortest contig or scaffold that cover 50% of assembly. However, Decapoda genomes present variable N50 values (Appendix A). The BUSCO completeness score, which can be independent of the contiguity of the genome, was also determined for each genome to assess the completeness of the assemblies (Table 1) [45]. Only the 20 genomes with a BUSCO completeness score of at least 25% were selected. Considering the low number and fragmentation status of available Decapoda genomes, a lower BUSCO score threshold than usually used was chosen to retain at least one genome in all infraorders that had genome assemblies. To obtain a broader perspective of the landscape of Decapoda REs compared to crustaceans, we added 6 non-Decapoda crustaceans (Table 1). This allowed us to see if Decapoda species have a different or similar trend in terms of the proportion of the individual repeat families, the presence/absence of RE families, and finally their evolutionary trajectories in comparison to six other Crustacea.

### 2.2. Identification and Annotation of Repetitive Elements

#### 2.2.1. Identification of Satellite DNA Families

For each species, a set of Illumina paired-end reads was randomly chosen in the SRA database (Table 1). Reads that mapped to the mitochondrial genome were discarded, and the remaining reads were sampled to represent 1.6% of estimated genome size. Genome size estimations were retrieved for all genomes, except for *Chionoecetes opilio* (Table 1). For this genome, all short paired-end reads corresponding to the assembly were downloaded and the genome size was estimated using KmerGenie version 1.7051 [65]. The sets of reads were then analysed using the TAREAN pipeline, Galaxy version 2.3.8.1 [66] (reads trimmed at 100 bp and default parameters) to compile each species-specific library of satellite elements.

#### 2.2.2. Construction of a Common Library of Repetitive Elements

*De novo* identification of repetitive elements in each genome was performed using RepeatModeler2 version 2.0.1 [38] with the LTRStruct option and default parameters. The LTRStruct option is an LTR structural discovery pipeline that allows a better identification of LTR elements by using LTR_Harvest and LTR_retriever.

All species-specific libraries of repetitive elements identified with RepeatModeler2 were renamed according to the RepBase version 26.05 [67] nomenclature, with the repeat family, a unique number for the family to distinguish the different sequences of the repeat, the 3-letter species name, the repeat class and family, and finally the complete species name. Similar renaming was applied to species-specific libraries of high-confidence satellites identified by the TAREAN pipeline, with the addition of a ‘tarean’ tag after the unique number.

All species-specific libraries of high-confidence satellites and repeats identified by the TAREAN pipeline and RepeatModeler2 were combined with the Arthropoda-specific subset of RepBase26.05 to form a single library (Figure 1). This library was then split into 2 sub-libraries. The first one corresponds to the known TEs and the second one represents unknown TEs, satellites, and simple repeats.

#### 2.2.3. Identification of Repetitive Elements

In order to annotate repetitive elements that are present in the 26 crustacean genomes, we used RepeatMasker version 4.1.2-p1 [39] following a two-step approach (Figure 1). First, we used RepeatMasker with the library of known TEs using the options -a -gccalc -excln -s -nolow to identify and mask TEs in genomic sequences. We then performed a second run of RepeatMasker (with -a -gccalc -excln -s options) on the previously masked genomes using the second library to identify unclassified TEs, satellite DNA, and simple repeats. The ProcessRepeats and buildSummary tools of RepeatMasker were then used to combine all results and produce a detailed summary of annotations.

#### 2.2.4. Statistical Analysis

In order to test for correlation between genome size, assembly size, repeats, or TE load (number of copies) or percentage, we used a linear regression model and the Spearman rank sum method with α = 0.005 using R package ggplot2 with lm method. A dendrogram was produced by calculating pairwise distances between repeat profiles (the pattern of presence and absence of repetitive elements) using hclust with the Euclidean method, and the heatmap was plotted using Orange3 [68]. The sequence divergence distribution was calculated as Kimura distances (rates of transitions and transversions) using the RepeatMasker tools “calcDivergenceFromAlign.pl” and “createRepeatLandscape.pl”.

## 3. Results and Discussion

### 3.1. Construction of Repetitive Elements Reference

To obtain a comprehensive view of REs in Decapoda and reduce the number of elements classified as “unknown”, we developed a standardized protocol to annotate TEs and satDNAs at the genomic level (see Methods and Figure 1). This pipeline integrates the consensus sequences of the Arthropoda section of the RepBase database and the *de novo* identification of REs in all species by a combination of RepeatModeler2 and the TAREAN pipeline, in order to generate an extensive library of consensus sequences. The TAREAN pipeline was used to specifically identify satDNAs. Due to their structure and high sequence homogeneity, satDNAs are extremely difficult to assemble and are often excluded from the assembly [12]. Therefore, we searched for satDNAs in Illumina raw reads paired-end sequences using the TAREAN pipeline to construct the “Satellite libraries”. Using the TAREAN pipeline, we retrieved between 0 and 43 satDNA families annotated as “High fidelity”, while RepeatModeler2 identified only 0 to 4 satDNA families (Table 2).

Using our newly developed pipeline, we identified between 3643 and 11,431 families of REs in the different assemblies, including between 7.25% and 33.37% of “unknown” sequences (Table 2). Unknown elements are repetitive sequences that could not be further classified. The lowest percentage of unknown elements is observed in Dendrobranchiata species. This might be explained by the presence of the annotated TEs of the Dendrobranchiata *Penaeus vannamei* in RepBase, allowing a better identification in closely related species.

All detected REs were renamed according to the RepBase nomenclature. In fact, the RE classification by Wicker et al. (2007) [35] is widely used, but new TEs have been characterized since the establishment of the classification in 2007, resulting in conflicts in TE databases. Kojima (2019) [37] improved the classification of the RepBase database [40], but TE annotations can differ between RepBase, RepeatModeler2 database, and DFAM due to capital letters or multiple naming of the same element, for example. A manual correction of repeat names was thus applied when needed in order to obtain a clear annotation.

All libraries generated by RepeatModeler2, the TAREAN pipeline, and RepBase were merged into a single library. This extensive database contains a total of 71,601 sequences including sequences from RepBase. Among these families, known TEs represent 31,579 sequences. With this new merged library, we considerably extended the number of annotated families compared to the RepBase database of Arthropoda REs. Indeed, RepBase provides consensus sequences of 13,906 repetitive elements in Arthropoda, including 109 satDNAs. These elements are distributed in 218 Arthropoda species and in Eukaryota or Metazoa common ancestors. However, only sixteen Crustacea and six Decapoda species are represented, with 1419 and 328 sequences, respectively. Moreover, most Decapoda sequences (320) are from a single species, *P. vannamei*, as repeats from other species have not been submitted to RepBase. This shows the lack of knowledge of REs in Decapoda species in established databases. Our work also extended the number of known satDNA families in Decapoda species, with 405 consensus sequences compared to the 109 present in RepBase. The new REs identified in this study are provided in Appendix A (Appendix A). Well-categorized REs have also been submitted to RepBase.

### 3.2. Annotation of Repetitive Elements in Decapoda Genomes

With our new extensive database, we performed two rounds of annotation using RepeatMasker. In the first round we only used known TEs in order to have a better characterization and reduce the proportion of unknown TEs, and in the second we used all the remaining REs. We identified between 6805 and 31,798 consensus RE sequences in the different assemblies (Table 2). This represents an increase of approximately 16,500 families on average in Decapoda compared to previous annotations and 6500 for the other Crustacea. Moreover, our standardized protocol successfully identified the type of REs that were previously unclassified for most species (now between 4.40% and 24.15%). This represents a considerable improvement over the results obtained with the widely used species-specific databases.

Taking into account all the satDNA families annotated in the genome with the merged library, we annotated between 11 and 109 different families (previously 10 to 40 using the species-specific strategy, Table 2). The Astacidea and Anomura infraorders have higher numbers of satDNA families, ranging from 92 to 109, except for *H. americanus* and *B. latro*. The latter two species have a number of satDNA families more similar to the other Decapoda species, with 61 and 59 satDNA consensus sequences, respectively. The large number of satDNA families detected in Astacidea and Anomura is in agreement with the 258 families detected in the crayfish *Pontastacus leptodactylus* [5]. The diversification of satDNA families in Astacidea and Anomura is remarkable compared to the observations in other species. For example, *Drosophila* species generally have less than ten different families in their genomes, and humans have nine [20,69]. However, a large number of satDNA repeats has already been found in Arthropoda, such as *Triatoma infestans* (42 families, genome size 1.4 Gb) [70], *Locusta migratoria* (62 families, genome size 6 Gb) [18], the morabine grasshoppers (129 families, genome size 5 Gb) [71], and the fish *Megaleporinus microcephalus* (164 families, assembly size 1.2 Gb) [72]. It should be noted that our results may still underestimate the real number of satDNA families, due to the fragmentation of available assemblies (Appendix A). In fact, some satDNA families identified by the TAREAN pipeline in Illumina reads were not retrieved in the genome assembly. It is likely that the missing satDNAs were contained in reads that were not included in the final assembly. However, the number of satDNAs remains consistent in each infraorder.

Interestingly, the number of RE families is correlated with both estimated genome size and assembly size (Table 1) with a Spearman rank correlation test of ρ = 0.83, *p*-value = 8.925 × 10^−8^ and ρ = 0.92, *p*-value = 1.146 × 10^−6^, respectively. The same correlation is observed with satDNA families, with Spearman rank correlation test of ρ = 0.84, *p*-value = 6.875 × 10^−8^ and ρ = 0.90, *p*-value = 3.83 × 10^−10^, respectively. This result reveals the importance of the diversification of RE families in larger genomes.

The strategy used in this study increases the knowledge of REs in Decapoda species and provides an extended library that can be used in future studies (Appendix A). Unfortunately, there are still a large number of unknown REs in some of the annotated genomes. A manual curation of the library would be necessary but was beyond the scope of this study. We also want to mention that, due to the high presence of REs, genome assemblies are often fragmented, preventing the exhaustive annotation of TEs that can be absent from the assemblies or split into two contigs. The study of Sproul et al. (2022) of more than 600 insect species showed the influence of sequencing technology on repeat detection, with long read assemblies containing 36% more repeats than short-read assemblies and a huge impact on LTR detection [73]. This is because assemblies based on long reads are often more contiguous [74,75]. In our case, most of the genomes were assembled using long reads or a combination of long and short reads, and short-read assemblies do not stand out concerning repeat content or diversification (Appendix A).

### 3.3. Proportion of Repetitive Elements in Decapoda Genomes

The RE proportions are variable both between and within phylogenetic clades of the analysed species. The proportion of REs in the studied Arthropoda genomes is above 40%. Exceptions are two Decapoda species, *C. quadricarinatus*, with the lowest contig N50, and *C. multidentata*, with the lowest BUSCO score. They present 38.73% and 39.02% of repeat content, respectively (Table 1, Figure 2, and Appendix A). The non-Decapoda *H. azteca* also presents fewer REs, with 26.12%, and is one of the genomes assembled with short reads only (Figure 2 and Appendix A), but given the fragmented status of these genomes, these percentages may underestimate the RE proportion. Compared to the Decapoda species, which have an average of 59.7% REs in their genomes, the non-Decapoda Crustacea analysed in this study exhibit a lower proportion of REs, with an average of 46.4%. However, it is important to note that *A. vulgare* stands out among the non-Decapoda studied, as it has a remarkably high percentage of repeats (76.26%). If *A. vulgare* is excluded, the average of REs in non-Decapoda is reduced to 40.4% and the difference is significant, with Wilcoxon *p*-value = 0.0074. Within Decapoda species, Anomura presents an especially high percentage of REs, with on average 73.6%. Indeed, the Anomura species *P. platypus* has the highest proportion of REs among the studied species with 78.89% (Figure 2). In contrast, the genome with the lowest percentage of repeats was the non-Decapoda *H. azteca* with 26.12%. Thus, the RE proportions were highly variable among the phylogenetic clades, as was the content of RE categories.

We also observed a variability in the content of REs within suborders. Among Decapoda, Dendrobranchiata exhibited half the amount of LINEs compared to Pleocyemata, with up to 35.3% in the Astacidea *C. destructor* (Figure 2). Dendrobranchiata was characterized by a high proportion of DNA transposons, for example in *A. vulgare,* with between 13% and 18% of DNA transposons. The Anomura infraorder has the highest percentage of LTRs, with more than 16%, and the Achelata *P. ornatus* has the lowest, with 3.24%. SINE elements were rare in all genomes, ranging from 0.02% in *H. Azteca* to 2.54% in *P. trituberculatus*. DIRS elements contribute less than 1% of the repeat content in almost all genomes. The main exception was *M. nipponense*, where DIRS represented 8.84%. This species also has the highest proportion of Penelope elements, with 5.18%. The infraorder with the second highest number of Penelope elements was Astacidea, with a mean of 2.3%. Unclassified elements were less frequent in the Dendrobranchiata suborder, with around 3.5%, probably because of the better characterization of REs in this suborder in the RepBase database, with the almost exclusive presence of annotations derived from *P. vannamei*. Therefore, more divergent species present a higher proportion of unclassified elements, such as *E. affinis* with 24.15%. The content variability suggests that the different suborders of the studied crustacean species have specific major REs present in their genomes.

According to RE studies of Decapoda species included in assembly publications, the proportion of REs varies from 8% to 82% [48,49,50,51,53,54,55,76,77,78,79,80,81,82,83,84]. Tan et al. (2020) annotated the repeatome of eight decapod species and estimated repetitive content between 27% and 50%, with the majority of the genomes having more LINEs, except for *P. vannamei*, which had more DNA transposons [54]. Compared to these studies, we annotated approximately 10% more repeats with our pipeline. For the *P. virginalis* genome, 8.8% of repetitive elements were retrieved in the assembly Pvir0.4 (GenBank accession: GCA_002838885.1) and 27.52% in the study of Tan et al. (2020) [54]. However, in the assembly DKFZ_Pvir_1.0 (GenBank accession: GCA_020271785.1), the new assembly version used in this study, we annotated 57.87% of repetitive elements [51,55]. In the assembly of *P. clarkii*, Xu et al. (2021) annotated 82.42% of repeats, while in our study, we observed only 71.26% (Figure 2). For the *P. platypus* genome, we observed similar overall results to Tang et al. (2021) (Figure 2) [56]. However, the percentages of LINEs and LTRs are increased by almost 10% each, while unknown TEs were reduced to 17%. The percentage of REs in *E. sinensis* was estimated at 40.5% and 61.42% in two different studies [54,85], while here we determined that repetitive elements represent 58.93% of the genome (Figure 2). Taken together, these results show that our method provides greater or equal proportion of REs but with a better characterization.

The Decapoda species studied here all presented high proportions of REs, ranging from 58% to 79% (Figure 2). They are in the upper range of what is generally observed in Arthropoda. Indeed, comparative studies carried out on arthropods (mainly based on insects) report highly variable proportions of TEs, ranging from 1% to 80% [73,86,87]. We expect even higher proportions of REs with the forthcoming sequencing of giant genomes in Decapoda or other Crustacea. Recently, the assembly of the Antarctic krill (belonging to a sister order of Decapoda) demonstrated that 92% of its genome is constituted of REs, 78% of them being TEs, indicating that Arthropoda can have an extremely high proportion of REs [88]. In terms of TE landscape, Decapoda presents only a few SINE elements, as for all Arthropoda (Figure 2). Previous studies in Dendrobranchiata species reported that the most abundant groups of repeats, disregarding simple sequence repeats, were DNA transposons or LINEs, with different results depending on the bioinformatic tools used [73,86,87]. Here, we showed that DNA transposons were the major subclass in all Dendrobranchiata species, followed by LINEs (Figure 2). This is similar to what is observed in most insect species, where DNA transposons are generally the major TE group present in genomes [73,86,87]. Interestingly, our results revealed a different situation in the studied Pleocyemata species, where LINE and LTR elements are more abundant (Figure 2). This can be compared to what is observed in some insect orders exhibiting a different TE composition: LTRs are more abundant in Diptera species, and Odonata and Orthoptera species are richer in LINE elements [73,86]. The change in the major type of REs between suborders suggests an altered strategy for genome stability maintenance and regulation of REs between suborders. Sproul et al. (2022) demonstrated that LINE-rich species lineages present many REs that are associated with protein-coding genes [73]. Such associations suggest consequences regarding phenotype evolution. The presence of a TE near a gene can lead to methylation changes. Indeed, it already has been shown that LINEs can serve as amplifiers for silencing away from the X-chromosome inactivation center, and LINEs and SINEs for gene imprinting [34,89]. The movement of a LINE, or other TE, to a new genomic locus, can thus have an impact on nearby gene expression, and ultimately reshape gene expression networks and impact genome evolution.

### 3.4. Correlation between Genome Size and Repetitive Elements

The 20 Decapoda species analysed in the present study have large differences in genome size estimations (1.6 Gb to 8.5 Gb). These differences were also evident in assembly sizes, although less pronounced (1 Gb to 4.8 Gb). The variability of the genome sizes raised the question of the contribution of REs to their host genome. After masking each genome, we calculated the load of REs, i.e., the number of copies of REs and TEs only, and the percentage of REs and TEs only. We then tested for a correlation between the aforementioned values and both assembly size and estimated genome size. The assembly size was positively correlated with both the load (ρ = 0.87, *p*-value = 1.864 × 10^−6^) and the percentage of TEs (ρ = 0.6, *p*-value = 1.48 × 10^−3^) (Figure 3A,B). The estimated genome size (Table 2) was positively correlated with the load of TEs (ρ = 0.62, *p*-value = 7.114 × 10^−4^), but there was no significant correlation with the percentage of TEs (ρ = 0.47, *p*-value = 1.421 × 10^−2^) (Figure 3C,D). Although the number of satDNA families was correlated with both assembly size and estimated genome size, when satDNA elements are included, the significance of the correlation between the load of REs and genome/assembly size is smaller (Appendix A). The correlations between the percentage of REs and both assembly and estimated genome size were not significant, with α = 0.005 (Appendix A).

For the first time in Decapoda species, a strong correlation is demonstrated between assembly size and load (number of copies) of TEs. This strong positive correlation reveals the impact of the number of TEs on the size of the assembly, with larger genomes associated with a higher presence of TEs. The percentage of TEs or REs is more often analysed than the load. In our study, the percentage of TEs was less significantly correlated with genome or assembly size than the load of TEs, and REs were not correlated with genome size. As in our study, Petersen et al. (2019) [86] found a positive correlation between the percentage of TEs and assembly size in arthropods, but they also found a positive correlation between the percentage of TEs and estimate size, which was not observed in our study. Moreover, Sproul et al. (2022) [73] found a positive correlation between the proportion of REs and assembly size in insects, which was not confirmed in our study. The differences between our results and the cited studies are likely due to the difficulties in assembling REs in large genomes such as Decapoda [73,86]. During assembly, REs can be excluded from the assembly even if they are present in the genome. It is therefore expected that REs are more correlated with assembly size than the estimated size. REs can also be fragmented and included in the assembly only partially, contributing to the load of REs in the genome but not to the percentage. This could explain the higher correlation coefficient observed for the load of REs in Decapoda genomes and highlights the usefulness of studying both percentage and load of REs in fragmented assemblies. The presence of fragmented REs is particularly true for satDNAs, which are often concatenated, since the assembler cannot define how many repetitions are present if they are not entirely covered by a long read. These difficulties in assembling satDNAs are particularly pronounced when assemblies are highly fragmented, as in this study, and could explain the decrease in or absence of the significance of the tests when including satDNAs. An improvement in genome contiguity could therefore affect inferences of correlation between REs and genome size. However, removing genomes of BUSCO score of less than 50% does not change conclusions on correlations between repeats and genome size.

### 3.5. Frequency of satDNA Families Occurrence

In Crustacea, and particularly in Decapoda, we annotated a large number of different satDNA families (Table 2) and evaluated the occurrence of each family in each genome (Figure 4). In each genome, the majority of satDNA families were detected one to nine times. Depending on the genomes, between one and thirty-four families appeared between 10 and 99 times. With nine out of the ninety-seven satDNA families repeated more than 1000 times, *P. clarkii* was the species with the highest number of highly repeated satDNA families. In contrast, five genomes do not have highly repeated satDNA families (more than 99 occurrences). Thus, although Decapoda has extremely large numbers of satDNA families (Table 2), only a few are predominant in each genome (Figure 4), as seen in several other studies [18,19,20]. The Decapoda and non-Decapoda species studied here are no exception. The Decapoda infraorders Astacidea and Anomura had the largest genome size estimation and assembly size (Table 1) and presented the largest numbers of families that were highly repeated in their genomes (Figure 4). They also tend to have the highest total number of families (Table 2). This suggests that satDNA is a key factor in explaining the huge variations in genome size observed in Decapoda.

### 3.6. Diversity of Repetitive Elements

To investigate the diversity of REs, we determined the number of copies (the load) of each superfamily of REs identified for each genome (Figure 5). With 67 superfamilies of TEs present in at least one species, the majority of the known superfamilies of REs were found in the investigated genomes, as seen in insects [86], and appear highly conserved across all the genomes (Figure 5). Among the studied Decapoda genomes, there was a clear pattern of high and low presence of repeat superfamilies, with only a few distinct variations between species by repeat suborder.

The load of REs of each superfamily was then used as a profile for each genome to construct the dendrogram by clustering of the RE profiles (Figure 5). This dendrogram mainly followed the currently known species phylogeny [4] except for *A. vulgare*, whose RE proportions and composition were more similar to Decapoda (Figure 2) and two Anomura species that were grouped with the Caridea. The genome of *A. vulgare* (1.6 Gb) was larger than the other Crustacea analysed in this study (238 Mb–1 Gb), with the highest percentage of repeats among the studied non-Decapoda crustacean species (Figure 2). This may explain why *A. vulgare* is clustered with Decapoda species and not with other crustaceans (Figure 5). Nevertheless, we could see a clear differentiation between Decapoda species and the other Crustacea that have a lower number and a distinct composition of REs, except for *A. vulgare*. Similarly, we could clearly distinguish Dendrobranchiata from Pleocyemata infraorders, with the presence of LINE *ingi* and SINE *MIR*. Within Pleocyemata, Caridea was also separated from the other Reptantia species, in agreement with the established phylogeny [4]. Many studies, including Petersen et al. (2019) [86], Sproul et al. (2022) [73], and Wu and Lu (2019) [87], based their RE analysis on already published phylogenetic trees. In our study, we clustered the repetitive profile of each genome and obtained a phylogenetic signal that respects the major classification (Figure 5) [1]. In fact, REs have been used recently as evidence for phylogenetic tree construction in plants, with RE abundance resolving species relationships in a similar manner to DNA sequences from plastid and nuclear ribosomal regions [90,91]. This can be explained by the capacity of some REs to have a high conservation and synteny within species [92,93,94]. This approach could therefore be used in the future to determine the phylogeny of non-model species using low-coverage, low-cost sequencing.

### 3.7. Sequence Divergence Distribution of Transposable Elements

The genetic distance between each annotated TE copy and the consensus sequence of the respective TE family was calculated using the Kimura 2P distance in order to analyse the sequence divergence distribution and approximate the age and intensity of duplication events (Figure 6). The distribution shows the genomic coverage of TE copies according to the percentage of divergence from their family consensus estimated using the Kimura 2P distance. A peak indicates that a large group of TE copies shares the same divergence to the consensus sequence and suggests a major expansion event of these elements. This event is more recent if the peak is located at a low Kimura 2P distance from the consensus, i.e., at a low percentage of divergence. At a high Kimura 2P distance, a wide peak can indicate that TE copies have undergone genetic drift or other processes, leading to high sequence divergence and suggesting an ancient expansion event.

In Dendrobranchiata, sequence divergence landscapes were similar for the five species (Figure 6). We observed two very similar peaks. The first one presented a larger number of LTRs and a smaller increase in LINE elements between 10% to 15% of divergence. The peak of LTRs was particularly high in *P. japonicus* and *P. indicus*. At the same time point, we observed an increasing amount of DNA transposons with the same distance to the consensus in *P. monodon*. A longer time ago, an augmentation of DNA transposons and LTR elements around 25% of divergence was shared by all species. This suggests that all the Dendrobranchiata shared the same old evolutionary events. The *P. monodon* genome was one of the few analysed Decapoda genomes showing a recent peak of SINE elements with the two *Procambarus* species. We would therefore expect to see a higher proportion of SINEs in *P. monodon* compared to other genomes. However, SINE elements were only slightly more abundant in this genome due to a higher presence of SINE MIR elements (Figure 2 and Figure 5). Interestingly, the content of repeats showed that DNA transposons are the most widespread among the suborder (Figure 2). However, the expansion of DNA transposons was older and more spread out over time (Figure 6). In contrast, the landscape and diversity of repeats showed a higher peak of LTR elements over time in the suborder compared to the other species, with Gypsy being the most abundant (Figure 5 and Figure 6). There were almost no sequences with low divergence. This quasi-absence of recent peaks in Dendrobranchiata suggests low activity of the TEs in recent times in these genomes (Figure 6).

The two Caridea species presented a different sequence divergence landscape (Figure 6). In *C. multidentata*, there was a recent peak of unknown elements between 5% to 10% of divergence. This peak could be caused by the expansion of one or several families of unknown TEs. We also observed that from high divergence, the fraction of the genome increased as the Kimura 2P distances decreased. This trend could be seen until the event at 5% to 10% of divergence. After this event, and more recently, the number of TEs with very low divergence decreased, with almost no TEs at 0% of divergence. This suggests that despite the peak of recently active unknown elements, TEs are not active anymore for this species. For *M. nipponense*, we observed two recent peaks at 1–4% and 10% of Kimura divergence corresponding to LINE, Penelope, and LTR elements for the first one and DIRS for the second one. We observed integrated virus expansion between 5% and 25% of divergence. This was in accordance with the diversification of repeats (Figure 5), where the *M. niponnense* genome was the Decapoda with the highest amount of integrated virus. The presence of sequences with little divergence from the consensus sequences suggests that TEs are active in this genome (Figure 6).

Within Astacidea, *H. americanus* has a different TE landscape compared to the other four species belonging to the infraorder (Figure 6). Indeed, the genome has a high peak at a divergence of 15% of unknown elements. Interestingly, we observed an ancient event concerning integrated viruses at 40% to 45% of Kimura 2P distance. The *H. americanus* genome was the only Decapoda genome studied here presenting this characteristic. Integrated virus could not be seen in the proportion of repeats because of their low presence in genomes and was included in the category “other REs” (Figure 2). Integrated virus in *H. americanus* sequences corresponds to the white spot syndrome virus (WSSV) [95], suggesting that *H. americanus* faced this virus a long time ago and these sequences were then propagated (Figure 6). Since WSSV is a worldwide threat to shrimps and potentially to many crustacean species, this interesting finding in a resistant species (i.e., *H. americanus*) could be important for future inferences into susceptibility/resistance to WSSV [96,97]. In the *H. americanus* genome, there was a clear increase in LINE, LTR, and DNA transposon coverage with a low percentage of divergence, which leads us to conclude that TEs are still active in this genome. TEs are also active in the *Procambarus* species, which has a similar landscape, with several elements at a low divergence and especially LINEs. We also observed an augmentation of Penelope and SINE elements at low divergence for both species. In *P. clarkii*, there was also a small peak at 10% of divergence of unknown elements. In contrast to the TEs in *C. quadricarinatus*, TEs seem to be active in *C. destructor*, with an increase in LINEs at low divergence. The expansion of LINEs in *C. quadricarinatus* was, instead, more ancient, at 6% to 10% of divergence.

In Brachyura, all genomes seemed to have active TEs, but the TE landscapes across the genomes of this infraorder differ from each other (Figure 6). In *P. trituberculatus*, the LINEs with no divergence from consensus sequences were three times more abundant than LINEs at 1% of divergence. These LINEs were in a very active phase in this genome. Penelope elements were also more abundant at 0% of divergence. The *C. sapidus* genome showed an almost constant increased coverage of TEs with lower divergence for all elements. However, we observed an increasing number of LTRs with no divergence and a decreasing number of LINEs and DNA transposons. The genome of *E. sinensis* was the only Brachyura genome presenting two peaks. The oldest one was at 15% of Kimura 2P distance and was caused by unknown elements. The latest event involved LINE, LTR, and unknown elements at divergences between 0% and 7%. Of the Brachyura, *C. opilio* had the least active TEs. We observed a large peak between 0% to 20% of divergence, where LINEs and LTRs increased. The proportion of DNA transposons also increased during this time, but at a lower coverage.

Concerning the last two infraorders, in Achelata, the *P. ornatus* genome has a middle age peak at 15% of divergence, corresponding to LTRs (Figure 6). There was also a recent and high peak, around 4–8% of divergence, caused by the expansion of LINE elements, with 2% of the genome being represented by LINEs that are 6% divergent. This suggests that LINEs were, until recently, highly transcriptionally active in the genome but are now inactive. The high presence of LINE elements was also visible when considering the proportion of repeats in the genome (Figure 2). In Anomura, the intragroup with the highest percentage of LTRs within Decapoda (Figure 2), *B. latro* and the *Paralithodes* species had very different landscapes. The *B. latro* genome seemed to have inactive TEs, with two peaks of LTRs and LINEs at 3% and 15% of Kimura 2P distance (Figure 6). On the other hand, *Paralithodes* species had highly active LINEs and LTRs, with 6.8% and 3.6% of LINE elements without divergence to consensus sequences in *P. platypus* and *P. camtschaticus,* respectively. Finally, for other crustaceans, the amount of unknown elements in their genomes was predominant, making the analysis of the divergence distribution of TEs in their genomes difficult to interpret (Appendix A).

A clear differentiation in sequence divergence distribution between Dendrobranchiata and Pleocyemata species was observed, as seen with the proportion and diversity of repeats (Figure 6). Indeed, Dendrobranchiata have more non-transcriptionally active TEs compared to the majority of Pleocyemata. Among all Pleocyemata species studied here, almost all have at least one or more types of active TEs. The expansion of a particular subfamily of RE increases genome plasticity and can indicate periods of rapid evolutionary changes [14,33]. This suggests that Pleocyemata genomes had a rapid evolution on a recent timescale. Genomes with recent accumulations of repeats present highly similar repeats or types of repeats that can be long (mostly LTRs and LINEs). These long repetitive regions are more difficult to assemble, and so repeat resolution during assembly is even more problematic [98]. Indeed, we could argue that a large number of the genomes studied presented recent accumulation of long REs. These long REs, being difficult to assemble, can be a possible explanation of assembly fragmentation. Moreover, species with larger genome sizes tend to have more transcriptionally active TEs, but also more REs.

## 4. Conclusions

In this study, we annotated repetitive elements in twenty Decapoda and six other Crustacea genome assemblies publicly available, using a new pipeline for the annotation of repetitive elements. We showed that repetitive elements constitute a large fraction of Decapoda genomes, with a highly variable content of REs both between and within infraorders of Decapoda. Additionally, our analysis indicates that in Decapoda, both the load of repetitive elements and the number of RE families are correlated with the assembly size of the genome. Moreover, larger genomes tend to have more active TEs (high proportion of sequences at 0% of divergence from their consensus), confirming the impact of REs in genome size expansion. We also demonstrated that, although the age distribution of TE superfamilies shows intra- and inter-lineage variation, the clustered RE profile reflects the phylogeny of the major groups analysed in this study. Compared to non-Decapoda Crustacea, Decapoda have a higher proportion and number of REs in their genome. Moreover, the pattern of RE families present in Decapoda is well-conserved across species. With our protocol, we showed that the combination of repeat libraries of all species provides an excellent tool to analyse content and diversification of repetitive elements with on average 8% more categorized elements. The new consensus sequences can improve the annotation of TEs in other Crustacea or Arthropoda species by increasing the number of consensuses for homology searches. We suggest using this two-step pipeline for all repeatome studies on non-model organisms that are often underrepresented in public databases. Our pipeline provides a baseline for future genomic analysis, producing standardized and reproducible analyses that will allow for much more rigorous and complete comparative analysis of repeats in non-model organisms.

## Figures and Tables

**Figure 1 genes-14-01627-f001:**
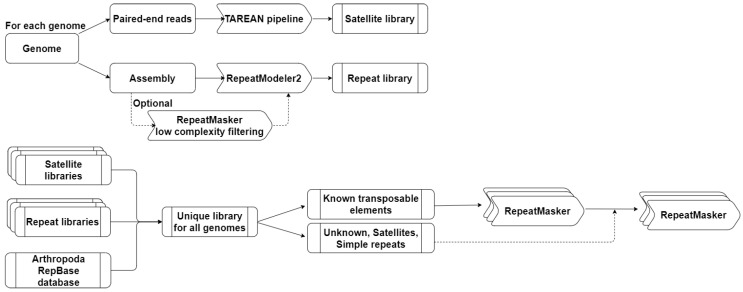
Standardized annotation protocol for repetitive elements developed in this study.

**Figure 2 genes-14-01627-f002:**
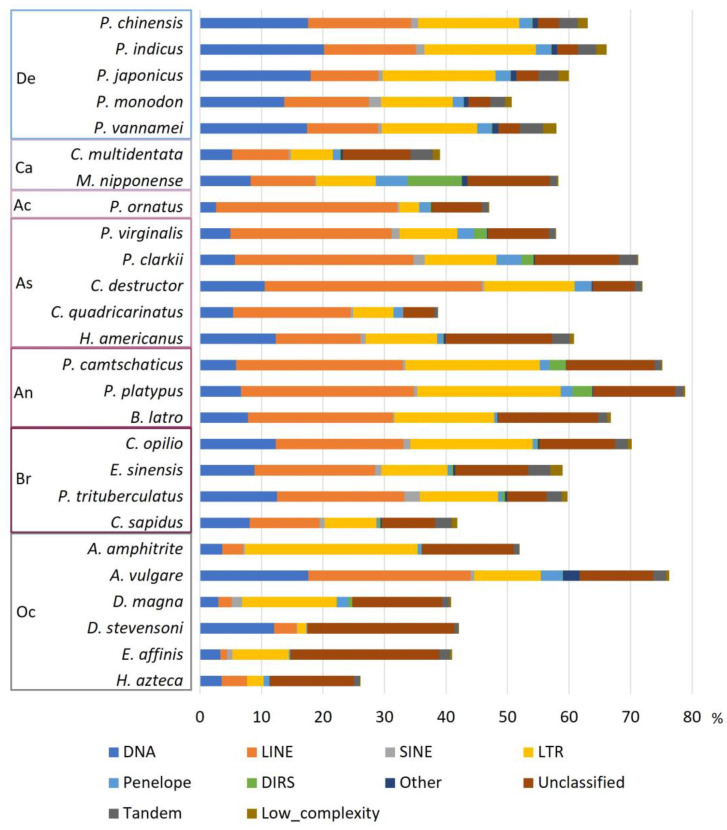
Proportion and content of repetitive elements in genomes. Percentage of repetitive elements in the genome by class of repetitive elements. De, Dendrobranchiata; Ca, Caridea; Ac, Achelata; As, Astacidea; An, Anomura; Br, Brachyura; Oc, other Crustacea.

**Figure 3 genes-14-01627-f003:**
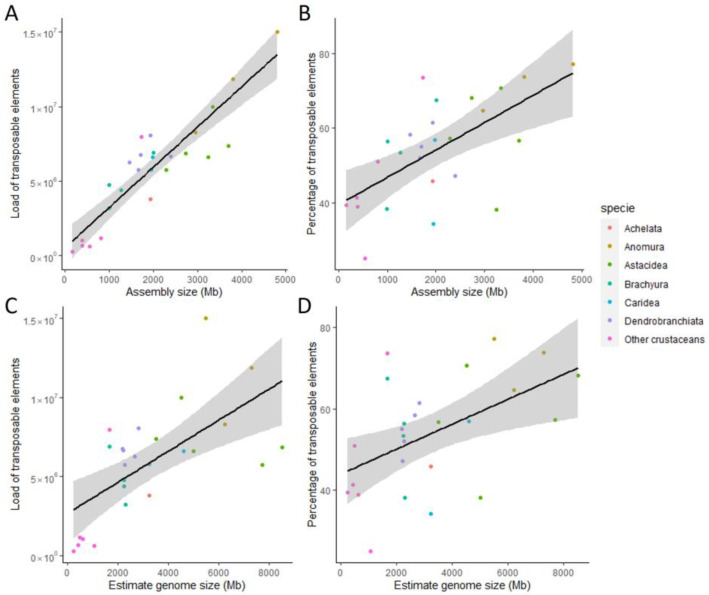
Correlation between genome size and TEs. Correlation plots between assembly or estimated genome size and load (number of copies) or percentage of TEs. Orders and suborders are indicated by different colours. (**A**). Correlation between assembly size and the load of TEs. Spearman rank correlation test: ρ = 0.87, *p*-value = 1.864 × 10^−6^. (**B**). Correlation between assembly size and the percentage of TEs. Spearman rank correlation test: ρ = 0.6, *p*-value = 1.48 × 10^−3^. (**C**). Correlation between estimated genome size and the load of TEs. Spearman rank correlation test: ρ = 0.62, *p*-value = 7.114 × 10^−4^. (**D**). Correlation between estimated genome size and the percentage of TEs. Spearman rank correlation test: ρ = 0.47, *p*-value =1.421 × 10^−2^.

**Figure 4 genes-14-01627-f004:**
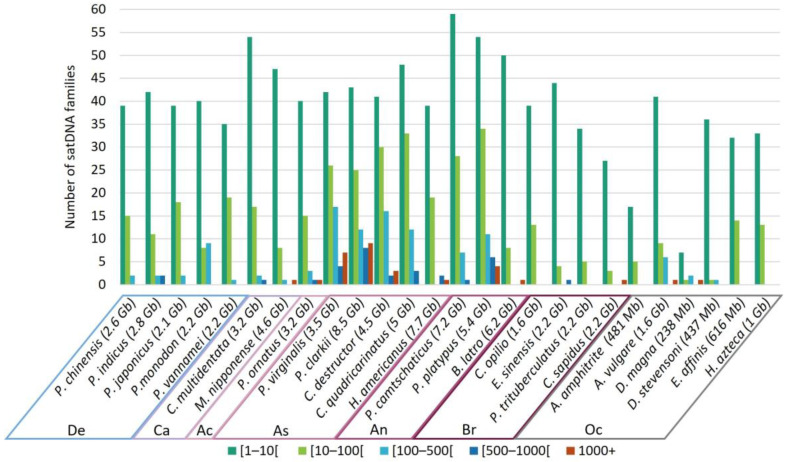
Distribution of satDNA families according to the number of occurrences in each genome. Low-frequency families (less than 10 occurrences) are indicated in dark green, while highly abundant families with more than 1000 occurrences are indicated in red. Number indicated for each species is the estimated genome size. De, Dendrobranchiata; Ca, Caridea; Ac, Achelata; As, Astacidea; An, Anomura; Br, Brachyura; Oc, other Crustacea.

**Figure 5 genes-14-01627-f005:**
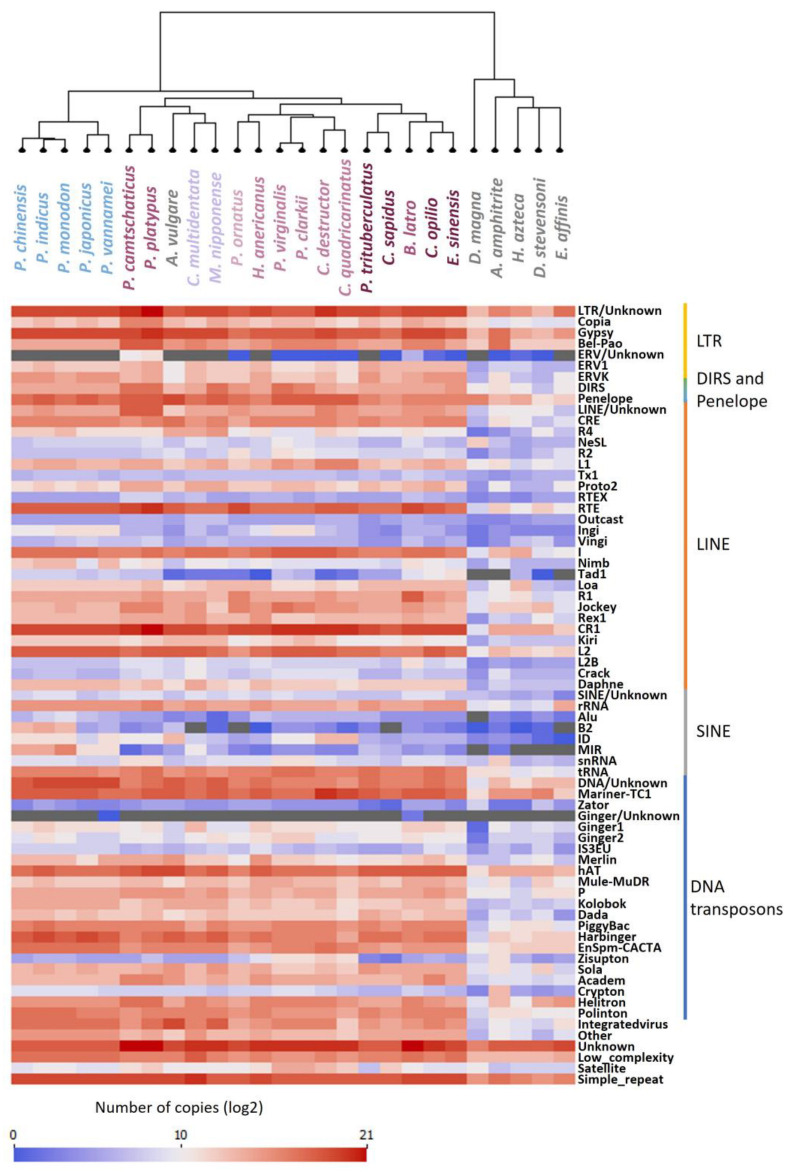
Diversity of repetitive elements. Log2 of the load of each family of repetitive elements identified for each genome was graduated between 0 (blue) and 21 (red). Gray colour indicates raw values of 0, before log2 transformation. The dendrogram was produced according to repeat profile by clustering.

**Figure 6 genes-14-01627-f006:**
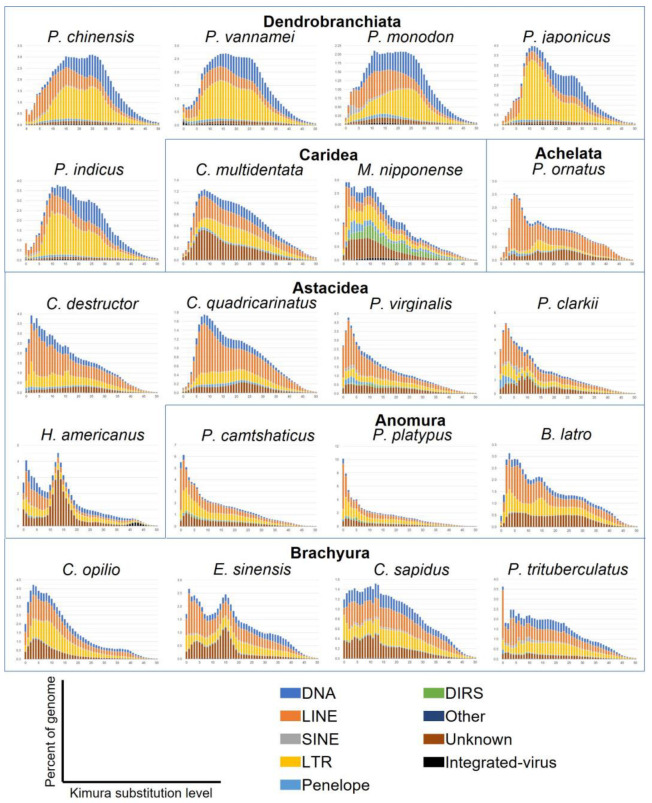
Sequence divergence distribution of TEs representing TE accumulation history based on Kimura 2P distance. Percentage of sequence divergence, or Kimura substitution level, is indicated on the *x*-axis. On the *y*-axis is the percentage of the genome occupied by each TE type; the scale is different for each genome depending on the percentage occupied. The TE type is indicated by the color chart.

**Table 1 genes-14-01627-t001:** Genomic datasets used in this study.

Suborder/Infraorder	Species	Assembly Access ID	Assembly Size (Mb)	BUSCO Completeness (%)	Paired-End Illumina Reads SRA Access ID	Estimate Genome Size (Mb)	Estimate Genome Size Reference
Dendrobranchiata	*Penaeus chinensis*	GCF019202785.1	1466	90.7	SRR13452153	2660	[46]
	*Penaeus indicus*	GCA018983055.1	1936	88.5	SRR12969543	2810	[47]
	*Penaeus japonicus*	GCF017312705.1	1705	96.6	DRR278744	2170	[47]
	*Penaeus monodon*	GCF015228065.1	2394	83.9	SRR11278066	2200	[47]
	*Penaeus vannamei*	GCF003789085.1	1664	84.8	SRR13661692	2270	[47]
Caridea	*Caridina multidentata*	GCA002091895.1	1949	25.2	DRR054559	3230	[48]
	*Macrobrachium nipponense*	GCA015104395.1	1985	41	SRR9026393	4600	[49]
Achelata	*Panulirus ornatus*	GCA018397875.1	1926	70	SSR13822589	3230	[50]
Astacidea	*Procambarus virginalis*	GCA020271785.1	3701	67	SRR12901906	3500	[51]
	*Procambarus clarkii*	GCF020424385.1	2735	94.3	SRR14457195	8500	[52]
	*Cherax destructor*	GCA009830355.1	3337	81.7	SRR10467055	4500	[53]
	*Cherax quadricarinatus*	GCA009761615.1	3237	69.9	SRR10484712	5000	[54]
	*Homarus americanus*	GCF018991925.1	2292	93	SRR12699166	7700	[55]
Anomura	*Paralithodes camtschaticus*	GCA018397895.1	3810	44.2	SRR13805857	7290	[50]
	*Paralithodes platypus*	GCA013283005.1	4805	71.7	SRR1145749	5490	[56]
	*Birgus latro*	GCA018397915.1	2959	57.7	SRR13816158	6220	[50]
Brachyura	*Chionoecetes opilio*	GCA016584305.1	2003	91	SRR11278230	1655	
	*Eriocheir sinensis*	GCA013436485.1	1272	92.6	SRR11971329	2230	[57]
	*Portunus trituberculatus*	GCF017591435.1	1005	93.5	SRR9964028	2250	[57]
	*Callinectes sapidus*	GCA020233015.1	998	90.4	SRR15834103	2290	[58]
Other Crustacea	*Amphibalanus Amphitrite* (Cirripedia)	GCA019059575.1	808	93.9	SRR9595623	481	[59]
	*Armadillidium vulgare* (Isopoda)	GCA004104545.1	1725	84.5	SRR8156178	1660	[60]
	*Daphnia magna* (Phyllopoda)	GCA020631705.2	161	98.6	SRR15012074	238	[61]
	*Darwinula stevensoni* (Podocopida)	GCA905338385.1	382	90.3	SRR8695251	437	[62]
	*Eurytemora affinis* (Copepoda)	GCA000591075.2	389	91	SRR2452640	616	[63]
	*Hyalella Azteca* (Amphipoda)	GCA000764305.4	551	93.8	SRR1556043	1050	[64]

**Table 2 genes-14-01627-t002:** Number of RE libraries identified and annotated using species-specific libraries or a merged library from all species. RMo—RepeatModeler2, Tp—TAREAN pipeline.

Suborder/Infraorder	Species	Ab Initio satDNA Families Identified		Number of Families Annotated Using RMo Species-Specific and Repbase as Library for Each Species			Number of Families Annotated Using Merged Libraries of RMo and Tp Libraries for All Species and Repbase		
		RMo	Tp	All RE Families	Percentage of Unknown	satDNA Only	All RE Families	Percentage of Unknown	Satdna Only
Dendrobranchiata	*P. chinensis*	1	7	7547	12.38%	24	22,702	3.44%	56
	*P. indicus*	1	2	8252	7.72%	30	24,237	3.40%	57
	*P. japonicus*	3	5	7693	7.25%	29	22,611	3.61%	59
	*P. monodon*	0	4	8647	9.28%	28	25,183	3.57%	57
	*P. vannamei*	0	3	7621	8.85%	30	23,240	3.49%	55
Caridea	*C. multidentata*	1	6	11,104	11.93%	38	28,065	11%	74
	*M. nipponense*	2	0	10,455	19.68%	38	26,021	13.42%	57
Achelata	*P. ornatus*	1	6	8850	21.13%	35	25,995	8.12%	60
Astacidea	*P. virginalis*	1	31	9213	28.26%	33	26,483	9.95%	96
	*P. clarkii*	2	39	8838	22.52%	34	26,051	13.67%	97
	*C. destructor*	4	24	10,391	14.10%	40	29,970	6.88%	92
	*C. quadricarinatus*	1	43	10,411	14.33%	35	26,966	4.99%	96
	*H. americanus*	1	2	9557	24.16%	35	27,873	17.29%	61
Anomura	*P. camtschaticus*	2	19	11,431	24.95%	33	30,169	14.36%	95
	*P. platypus*	0	36	11,332	32.76%	34	31,798	13.27%	109
	*B. latro*	1	2	11,053	25.48%	37	31,207	16.30%	59
Brachyura	*C. opilio*	0	0	10,400	22.89%	29	26,561	12.26%	52
	*E. sinensis*	1	0	8486	20.74%	29	23,937	11.82%	49
	*P. trituberculatus*	0	0	7399	12.28%	20	21,070	6.42%	39
	*C. sapidus*	0	2	6911	13.68%	18	19,041	8.68%	31
Other Crustacea	*A. Amphitrite* (Cirripedia)	1	1	6717	27.06%	14	11,969	14.90%	22
	*A. vulgare* (Isopoda)	0	13	9431	17.40%	27	19,098	11.91%	47
	*D. magna* (Phyllopoda)	2	3	3643	17.90%	10	6805	14.63%	11
	*D. stevensoni* (Podocopida)	1	2	9762	25.59%	22	17,339	23.89%	38
	*E. affinis* (Copepoda)	1	8	6069	33.37%	32	13,334	24.15%	46
	*H. Azteca* (Amphipoda)	1	10	6851	16.21%	28	14,424	13.69%	46

## Data Availability

In this study, we generated a library of repetitive elements in crustacean species. Elements fully categorized were submitted to Repbase. The library of new repetitive elements found during this study is also provided in Appendix A.

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
