# Peer review of "Abundance and Diversification of Repetitive Elements in Decapoda Genomes"

_genes, 2023, doi:10.3390/genes14081627_

Round 1
Reviewer 1 Report
This work establishes a novel pipeline to detect repetitive elements in DNA sequences, as well as their important functions. Further explores the relationship between transposable elements and genome size. However, the research did not interpret transposable elements thoroughly, some expressions were not clear. Moreover, there are some mistakes in formats and English expressions.
Please correct the article carefully for having better presentation of transposable elements.
Main question:
1. Page 2, Line 89: You should explain the difference between LINE and SINE retrotransposons separately.
2. Page 7, Line 210: Why most decapods come from P. vannamei species?
3. Page 10, Line 334: You should explain what “SSR” means.
4. Page 12, Line 379: As with Figure 3, you should explain what “the load” represents and what the purpose is of graphing “the load of TEs” separately from “the percentage of TEs”.
5. Page 15, Line 460: Why is the peak said to indicate that TEs have undergone an evolutionary event, simply because of the expansion of TEs elements?
6. Page 16, Line 467: What is the meaning of “sequence convergence” and what is the significance of different “sequence convergence” values on the x-axis?
7. Page 17, Line 488: Why low sequence divergence means TEs are not transcriptionally active in the genome?
8. You used 20 Decapoda and 6 other crustaceans to come up with a characterization of Decapoda, so what is the distinct evolutionary trajectories of the Decapoda genome in terms of repetitive elements?
The format of references needs to be checked and unified.
Author Response
We thank reviewer 1 for her/his constructive criticism on our manuscript genes-2521526 and believe this has improved the article. We have answered all queries below and modified the relevant sections in the manuscript. In addition, the entire text has now been proofread by a native English speaker. We hope this revised version will meet the standards of the journal and the expectations of the readership.
Comments and Suggestions for Authors
This work establishes a novel pipeline to detect repetitive elements in DNA sequences, as well as their important functions. Further explores the relationship between transposable elements and genome size. However, the research did not interpret transposable elements thoroughly, some expressions were not clear. Moreover, there are some mistakes in formats and English expressions. Please correct the article carefully for having better presentation of transposable elements.
The entire text has now been proofread by a native English speaker.
Page 2, Line 89: You should explain the difference between LINE and SINE retrotransposons separately.
Done as suggested. The differences between LINE and SINE are now explained at line 91: “Finally, LINEs (long interspersed nuclear elements) and SINEs (short interspersed nuclear elements) are retrotransposons that do not have terminal repeats but a polyA tail at the 3’ end. Unlike LINEs, SINEs evolved from non-coding RNA genes and are not autonomous.”
Page 7, Line 210: Why most decapods come from P. vannamei species?
Improved as suggested. The sequences referred to here only concern the RepBase database and not those identified in the study. This has been specified at line 226: “Moreover, most Decapoda sequences (320) are from a single species, P. vannamei, as repeats from other species have not been submitted to RepBase.”
Page 10, Line 334: You should explain what “SSR” means.
Done as suggested. The acronym SSR has been replaced by “simple sequence repeats” at line 363:” Previous studies in Dendrobranchiata species reported that the most abundant groups of repeats, disregarding simple sequence repeats, were DNA transposons or LINEs with different results depending on the bioinformatic tools used [53, 74, 75].”
Page 12, Line 379: As with Figure 3, you should explain what “the load” represents and what the purpose is of graphing “the load of TEs” separately from “the percentage of TEs”.
Modified as suggested. The term load (number of copies) is now explained in the main text and in the figure legend, as well as the purpose of plotting the load of repetitive elements.
Line 402: “Correlation plot between assembly or estimated genome size and load (number of copies) or percentage of TEs.”
Line 410: “For the first time in Decapoda species, a strong correlation is demonstrated between assembly size and load (number of copies) of TEs.”
Line 425: “During assembly REs can be excluded from the assembly even if they are present in the genome. It is therefore expected that REs are more correlated with assembly size than the estimated size. REs can also be fragmented and included in the assembly only partially, contributing to the load of REs in the genome but not to the percentage. This could explain the higher correlation coefficient observed for the load of REs in Decapoda genomes and highlights the usefulness of studying both percentage and load of REs in fragmented assemblies.”
Page 15, Line 460: Why is the peak said to indicate that TEs have undergone an evolutionary event, simply because of the expansion of TEs elements?
Improved as suggested. A peak can indicate an expansion of TEs that led to the divergence of TEs from their consensus. Line 510: “A peak indicates that a large group of TE copies share the same divergence to the consensus sequence and suggests a major expansion event of these elements.”
Page 16, Line 467: What is the meaning of “sequence convergence” and what is the significance of different “sequence convergence” values on the x-axis?
Improved as suggested. The sequence divergence is the percentage of substitutions of the TE from the consensus.
Line 512: “This event is more recent if the peak is located at a low Kimura 2P distance from the consensus, i.e., at a low percentage of divergence.”
Line 519: “Percentage of sequence divergence, or Kimura substitution level, is indicated on the x-axis.”
Page 17, Line 488: Why low sequence divergence means TEs are not transcriptionally active in the genome?
Improved as suggested. The presence of a peak of TEs at really low sequence divergence indicates that expansion of these TEs is recent. Therefore, TEs exhibit high identity because they did not have time to accumulate mutations. Taken together, a high abundance of lowly divergent TEs can only be achieved if TEs are actively propagating in the genome.
Line 512: “This event is more recent if the peak is located at a low Kimura 2P distance, i.e., at a low percentage of divergence.”
Line 540: “There were almost no sequences with low divergence. This quasi-absence of recent peaks in Dendrobranchiata suggests low activity of the TEs in recent times in these genomes (Figure 6).”
You used 20 Decapoda and 6 other crustaceans to come up with a characterization of Decapoda, so what is the distinct evolutionary trajectories of the Decapoda genome in terms of repetitive elements?
Improved as suggested. The differences between Decapoda and non-Decapoda is now stated in the conclusion at line 642: “Compared to non-Decapoda crustaceans, Decapoda have a higher proportion and number of REs in their genome. Moreover, the pattern of REs families present in Decapoda is well conserved across species.“
Comments on the Quality of English Language
The format of references needs to be checked and unified.
Done as suggested. The references are now unified.
Reviewer 2 Report
Abundance and diversification of repetitive elements in Decapa genomes
This paper is describing the developed a new standardized pipeline to annotate repetitive elements using 20 Decapoda and 6 other crustacean genomes. With this approach the authors have identified 10% more repetitive elements than standard pipelines. Data generated in this study provide baseline information for further comparative analyses in the genome of Decapods.
Abstract.
Line 22. Change satDNA for satellite DNA
Introduction
Line 100 delete “and many more”
Line 116. Please describe the criteria for choosing genomes with 25% of completeness were selected?. Non-model genome projects commonly report BUSCO scores ranging from 50% up to 95% complete, depending on the challenge posed by the species’ biology (e.g., genome size, amount of repetitive elements) and its taxonomic position.
See references
loannidis P, Simao FA, Waterhouse RM et al (2017) Genomic features of the Damselfly Calopteryx splendens representing a Sister Clade to most insect orders. Genome Biol Evol 9:415–430. https://doi.org/10.1093/gbe/evx006
Holt C, Campbell M, Keays DA et al (2018) Improved genome assembly and annotation for the rock pigeon (Columba livia). G3 Genes Genomes Genet 8:1391–1398. https://doi.org/10.1534/g3.117.300443
Line 117-118 Please describe fully this section, “ To have a broader perspective of Decapoda REs landscape compared to Crustaceans, we added other crustacean non-Decapoda”, what was the reason for including the crustacean non-decapoda?,
Comment: Running BUSCO provides high-quality gene model training data that can greatly improve genome annotation procedures. Gene prediction remains a challenging procedure, especially in the absence of supporting evidence such as native transcripts or homologs from close species.
Results and discussion
Lines 242-245. It is to note that our results may still underestimate the real number of satDNA families due to the fragmentation of available assemblies. In fact, some satDNA families identified by the TAREAN pipeline in Illumina reads weren’t retrieved in the genome assembly. It is likely that the missing satDNAs were contained in reads that were not included in the final assembly.
Comment: This is true, it is possible for the authors to comment on putative alternatives?, For example, in this study, the authors used only BUSCO. The primary goal of the BUSCO tool is to allow evaluation, comparison, and reevaluation of assemblies and annotations. It is suggested to discuss if by using NG50 or N50 that are not the same but can be complementary when evaluating genomes. N50 is the first approach used to evaluate the presence of fragmented segments.
Lines 263-264. In our case, most of the genomes were assembled using long reads or a combination of long and short reads, and short read assemblies do not stand out concerning repeat content or diversification. Please add the appropriate references or refer to table 1.
Lines 270-272. The proportion of REs in studied Arthropoda genomes is above 40%. Exceptions are two Decapoda species, Cherax quadricarinatus and Caridina multidentata, with respectively 38.73% and 39.02% of repeat content, and the non-Decapoda Hyallela azteca with 26.12% (Figure 2).
Comment: These results are because the repeat content are below 50%. One suggestion is to eliminate these data from the main body of the Ms and add the information as supplementary. I would not suggest to eliminate the information because we urgently need to characterize the unclassified and other boxes from non-model organisms.
Lines 289-296. Please add appropriate references in this segment.
Lines 385-393. Please edit this paragraph, in the current way it is difficult to follow. “This can be explained by the fragmentation of the genomes analysed due to the difficulties to assemble REs: REs can either be excluded from the assembly although present in the genome and cannot be annotated, or they can be fragmented indicating that a part of the RE is not included in the assembly and so can contribute to the load of REs in the genome but not to the percentage. This is the case for satDNAs that are often concatenated since the assembler cannot define how much repetitions are present if they are not entirely covered by a long read. This explains the decrease or absence of the significance of the tests when including satDNAs.
It is therefore expected that the load is rather correlated with the assembly size than with the estimated size”. Would this explain the lack of correlation analyses of this study with respect to the others?, what would be the biological explanation of the weak or strong correlation of the estimated size of the genome?, In this case, the authors are using assembled genomes based on BUSCO it is correct?.}
Lines 395-345. Please add appropriate references to this section.
Lines 455.510. Please add appropriate references to this section.
Line 511. Integrated virus in H. americanus sequences correspond to the white spot syndrome virus. This is an interesting topic to discuss because in H. americanus WSSV is not highly pathogenic as for Penaeus vannamei.
Line 536 change “lesser” by “lower”
Conclusion
This section needs some edition, please reduce the length of this section.
Minor corrections
Author Response
We appreciate reviewer 2 taking the time to read over our work, genes-2521526. Please find our responses to the reviewer's remarks below. We have clarified the portions that were requested to be modified from our previous version and addressed any questions. We also enhanced the overall English expressions in the manuscript. We thank the reviewers for constructive criticism and believe this has improved the text. We genuinely hope that our revised paper will satisfy the journal's requirements as well as the readership's expectations.
Comments and Suggestions for Authors
Abundance and diversification of repetitive elements in Decapa genomes
This paper is describing the developed a new standardized pipeline to annotate repetitive elements using 20 Decapoda and 6 other crustacean genomes. With this approach the authors have identified 10% more repetitive elements than standard pipelines. Data generated in this study provide baseline information for further comparative analyses in the genome of Decapods.
Abstract.
Line 22. Change satDNA for satellite DNA
The text has been changed in accordance with the suggestion.
Introduction
Line 100 delete “and many more”
This has been done as suggested.
Line 116. Please describe the criteria for choosing genomes with 25% of completeness were selected?. Non-model genome projects commonly report BUSCO scores ranging from 50% up to 95% complete, depending on the challenge posed by the species’ biology (e.g., genome size, amount of repetitive elements) and its taxonomic position.
Improved as suggested. The criteria of 25% of BUSCO score is now explained at line 122: “Contig and scaffold N50 are useful values to estimate the contiguity of the genome by indicating the length of the shortest contig or scaffold that cover 50 % of assembly. However, Decapoda genomes present variable N50 values (Table S1). The BUSCO completeness score, that can be independent of the contiguity of the genome, was also determined for each genome to assess the completeness of the assemblies (Table 1) [45]. Only the 20 genomes with a BUSCO completeness score of at least 25% were selected (Table 1). Considering the low number and fragmentation status of available Decapoda genomes, a lower BUSCO score threshold than usually used was chosen to retain at least one genome in all infraorders that had genome assemblies.”
We are aware that this criteria does not meet the gold standard in research, and therefore the conclusions about genomes with low BUSCO score or low N50 values are now nuanced in the Results and Discussion section at line 290: “The proportion of REs in studied Arthropoda genomes is above 40%. Exceptions are two Decapoda species, Cherax quadricarinatus, with the lowest contig N50, and Caridina multi-dentata, with the lowest BUSCO score. They present respectively 38.73% and 39.02% of repeat content (Table 1, Figure 2, and Table S1) but given the fragmented status of these genomes, these percentages may underestimate the RE proportion. The non-Decapoda Hyallela azteca present also less than 40% of REs, with 26.12%, and is one of the genomes assembled with short reads only (Figure 2 and Table S1).”
For this study, we removed one genome at 0.2% and one at 12.7% of BUSCO completeness. Moreover, removing genomes with BUSCO score below 50% did not affect conclusions about the correlations between repeats and genome size (line 441), dendrogram produced by the clustering or higher identification rate of repetitive elements when implementing our pipeline. On the contrary, we showcase that even the low quality genome assemblies can be used to perform the analysis of repeat landscapes of the non-model species. Lastly, utilisation of the incomplete genomic data to analyse repetitive proportion of the Eucaryotic genomes was previously proven and validated in the multiple studies that successfully utilised genome skimming data (ex. RepeatExplorer pipeline).
See references
loannidis P, Simao FA, Waterhouse RM et al (2017) Genomic features of the Damselfly Calopteryx splendens representing a Sister Clade to most insect orders. Genome Biol Evol 9:415–430. https://doi.org/10.1093/gbe/evx006
Holt C, Campbell M, Keays DA et al (2018) Improved genome assembly and annotation for the rock pigeon (Columba livia). G3 Genes Genomes Genet 8:1391–1398. https://doi.org/10.1534/g3.117.300443
Thank you for this suggested reading. Holt C. et al., (2018) is now cited as reference 45.
Line 117-118 Please describe fully this section, “To have a broader perspective of Decapoda REs landscape compared to Crustaceans, we added other crustacean non-Decapoda”, what was the reason for including the crustacean non-decapoda?
Improved as suggested. The rational of including non-Decapoda crustaceans is now provided in line 132: “This allowed us to see if Decapoda species have different or similar trend in terms of proportion of the individual repeat families, and their presence/absence of REs families and finally their evolutionary trajectories in other non-Decapod Crustaceans.”
Running BUSCO provides high-quality gene model training data that can greatly improve genome annotation procedures. Gene prediction remains a challenging procedure, especially in the absence of supporting evidence such as native transcripts or homologs from close species.
We agree with the reviewer that gene prediction and annotation remain challenging, especially in such fragmented genomes. However, gene prediction is not directly addressed in this manuscript.
Results and discussion
Lines 242-245. It is to note that our results may still underestimate the real number of satDNA families due to the fragmentation of available assemblies. In fact, some satDNA families identified by the TAREAN pipeline in Illumina reads weren’t retrieved in the genome assembly. It is likely that the missing satDNAs were contained in reads that were not included in the final assembly.
Comment: This is true, it is possible for the authors to comment on putative alternatives?, For example, in this study, the authors used only BUSCO. The primary goal of the BUSCO tool is to allow evaluation, comparison, and reevaluation of assemblies and annotations. It is suggested to discuss if by using NG50 or N50 that are not the same but can be complementary when evaluating genomes. N50 is the first approach used to evaluate the presence of fragmented segments.
As pointed out by the reviewer, N50 (and NG50) are complementary to BUSCO score, and we have indicated these metrics of genome contiguity in the supplementary materials as well as the type of reads used for sequencing. We also discussed the results obtained for genomes with a low BUSCO score and/or a low N50. Nonetheless, neither BUSCO nor N50 are directly measuring the quality of the repetitive proportion of the assembly. BUSCO is intrinsically limited because it is focused on the identification of the single-copy orthologs, while repetitive elements are multicopy. Therefore, the repeat profiles (landscapes) are visible and interpretable even in the low-quality genomic assemblies (as shown in our study). We have indicated contig and scaffolds N50 values in the supplementary materials as well as the type of reads used for sequencing. We also discussed the results obtained for genomes with a low BUSCO score and/or a low N50. (see below response for lines 270-272)
Lines 263-264. In our case, most of the genomes were assembled using long reads or a combination of long and short reads, and short read assemblies do not stand out concerning repeat content or diversification. Please add the appropriate references or refer to table 1.
Done as suggested. The sentence now refers to table S1, where the sequencing strategy used can be found.
Lines 270-272. The proportion of REs in studied Arthropoda genomes is above 40%. Exceptions are two Decapoda species, Cherax quadricarinatus and Caridina multidentata, with respectively 38.73% and 39.02% of repeat content, and the non-Decapoda Hyallela azteca with 26.12% (Figure 2).
Comment: These results are because the repeat content are below 50%. One suggestion is to eliminate these data from the main body of the Ms and add the information as supplementary. I would not suggest to eliminate the information because we urgently need to characterize the unclassified and other boxes from non-model organisms.
We agree that information on repetitive elements in these non-model organisms is urgently needed and this is why we retained as much data as possible. However, we recognise that it is important to avoid over-interpretation for the most fragmented genomes. We have therefore modified the text (line 291) to qualify the results obtained for these genomes. “Exceptions are two Decapoda species, Cherax quadricarinatus, with the lowest contig N50, and Caridina multidentata, with the lowest BUSCO score. They present respectively 38.73% and 39.02% of repeat content (Table 1, Figure 2, and Table S1). The non-Decapoda Hyallela azteca also presents less than 40% of REs, with 26.12%, and is one of the genomes assembled with short reads only (Figure 2 and Table S1) but given the fragmented status of these genomes, these percentages may underestimate the RE proportion. The non-Decapoda Hyallela azteca present also less than 40% of REs, with 26.12%, and is one of the genomes assembled with short reads only (Figure 2 and Table S1).
Lines 289-296. Please add appropriate references in this segment.
Done as suggested. References to respective figures are now added in the segment.
Lines 385-393. Please edit this paragraph, in the current way it is difficult to follow. “This can be explained by the fragmentation of the genomes analysed due to the difficulties to assemble REs: REs can either be excluded from the assembly although present in the genome and cannot be annotated, or they can be fragmented indicating that a part of the RE is not included in the assembly and so can contribute to the load of REs in the genome but not to the percentage. This is the case for satDNAs that are often concatenated since the assembler cannot define how much repetitions are present if they are not entirely covered by a long read. This explains the decrease or absence of the significance of the tests when including satDNAs.
It is therefore expected that the load is rather correlated with the assembly size than with the estimated size”. Would this explain the lack of correlation analyses of this study with respect to the others?, what would be the biological explanation of the weak or strong correlation of the estimated size of the genome?, In this case, the authors are using assembled genomes based on BUSCO it is correct?
Edited as suggested. This paragraph was modified for readability. Line 423: “The differences between our results and the cited studies are likely due to the difficulties in assembling REs in large genomes such as Decapoda [54, 75]. During assembly, REs can be excluded from the assembly even if present in the genome. It is therefore expected that REs are more correlated with assembly size than the estimated size. REs can also be fragmented and included in the assembly only partially, contributing to the load of REs in the genome but not to the percentage. This could explain the greater correlation observed for the load of REs in Decapoda genomes and highlights the usefulness of studying both percentage and load in fragmented assemblies. Presence of fragmented REs is particularly true for satDNAs which are often concatenated, since the assembler cannot define how many repetitions are present if they are not entirely covered by a long read. These difficulties in assembling satDNAs are particularly pronounced when assemblies are highly fragmented as in this study and could explain the decrease or absence of the significance of the tests when including satDNAs. An improvement in genome contiguity could therefore affect inferences of correlation between REs and genome size.”
Line 412: “This strong positive correlation reveals the impact of the number of TEs on the size of the assembly, with larger genomes associated with a higher presence of TEs”
We have now included additional metrics of contiguity, such as N50, in supplementary tables.
Lines 395-345. Please add appropriate references to this section.
Done as suggested. References to respective figures are now added in the section.
Lines 455.510. Please add appropriate references to this section.
Done as suggested. References to respective figures are now added in the section.
Line 511. Integrated virus in H. americanus sequences correspond to the white spot syndrome virus. This is an interesting topic to discuss because in H. americanus WSSV is not highly pathogenic as for Penaeus vannamei.
Improved as suggested. The WSSV is discussed at line 569: “Since WSSV is a worldwide threat to shrimps and potentially to many crustacean species, this interesting finding in a resistant species (i.e., H. americanus) could be important for future inferences into susceptibility/resistance to WSSV [85, 86].”
Line 536 change “lesser” by “lower”
This has been done as suggested.
Conclusion
This section needs some edition, please reduce the length of this section.
The length of the conclusion has been reduced.
Comments on the Quality of English Language
Minor corrections
Done as suggested. The entire text has now been proofread by a native English speaker.
Round 2
Reviewer 1 Report
Accept in present form.
Reviewer 2 Report
Thank you for addressing my previous queries. In this revised version there are no further queries.
Minor English corrections are suggested